# A Novel and Simplified Extrinsic Calibration of 2D Laser Rangefinder and Depth Camera

**Wei Zhou** [1,2,†]**, Hailun Chen** [1,3,†]**, Zhenlin Jin** [2]**, Qiyang Zuo** [1,3]**, Yaohui Xu** [1,3] **and Kai He** [1,3,*]

1   Shenzhen Institute of Advanced Technology, Chinese Academy of Sciences, Shenzhen 518055, China; wei.zhou1@siat.ac.cn (W.Z.); hl.chen@siat.ac.cn (H.C.); qy.zuo@siat.ac.cn (Q.Z.); yh.xu@siat.ac.cn (Y.X.)
2   School of Mechanical Engineering, Yanshan University, Qinhuangdao 066004, China; zljin@ysu.edu.cn
3   Shenzhen Key Laboratory of Precision Engineering, Shenzhen 518055, China
*   Correspondence: kai.he@siat.ac.cn
†   These authors contributed equally to this work.

**Abstract:** It is too difficult to directly obtain the correspondence features between the two-dimensional (2D) laser-range-finder (LRF) scan point and camera depth point cloud, which leads to a cumbersome calibration process and low calibration accuracy. To address the problem, we propose a calibration method to construct point-line constraint relations between 2D LRF and depth camera observational features by using a specific calibration board. Through the observation of two different poses, we construct the hyperstatic equations group based on point-line constraints and solve the coordinate transformation parameters of 2D LRF and depth camera by the least square (LSQ) method. According to the calibration error and threshold, the number of observation and the observation pose are adjusted adaptively. After experimental verification and comparison with existing methods, the method proposed in this paper easily and efficiently solves the problem of the joint calibration of the 2D LRF and depth camera, and well meets the application requirements of multi-sensor fusion for mobile robots.

**Keywords:** two-dimensional laser-range-finder; depth camera; extrinsic calibration; data fusion





## 1. Introduction

With the rapid development of sensor technology and computer vision technology, laser-range-finder (LRF) and cameras have become indispensable sensors for autonomous driving, mobile robots and other fields [1]. Two-dimensional (2D) LRF is commonly used to measure depth information in a single plane due to its high precision, light weight and low power consumption. The camera acquires rich information, such as color and texture, but it is sensitive to light and weather, resulting in its poor stability. On the other hand, it is difficult for the camera to measure depth directly over long distances. Therefore, laser vision fusion plays an important role in robot self-localization [2,3], environmental perception [4], target tracking [5], and path planning [6].

To integrate data information from 2D LRF and depth cameras, the relative positional relationship between the two sensors needs to be precisely known [7]. This is a classical extrinsic calibration problem, where the objective is to determine the conversion relationship between two coordinate systems. In contrast to 3D LRF, which identifies different features, 2D LRF only measures depth information in a single plane, and it is difficult for the camera to see the plane scanned by 2D LRF, which makes extrinsic calibration for 2D LRF and cameras more challenging. Therefore, additional constraints must be used to find the correspondence between the 2D LRF and the camera.

There has been a large amount of research work on the extrinsic calibration of 2D LRF and cameras, which is divided into two categories: target-based calibration and non-target calibration. References [7–24] are target-based calibration. Zhang and Pless [8] proposed a method by using point constraints on a plane, but only two degrees of freedom are

constrained in a single observation, which required a large number of different observations to ensure accuracy. Vasconcelos et al. [9] proposed to solve the problem in [8] by forming a perspective three-point (P3P) problem. Zhou [10] further proposed an algebraic method for extrinsic calibration. Both methods in [9,10] required multiple observations and suffered from multi-solution problems. Kaiser et al. [11] proposed a calibration algorithm such that the problem of rigid displacement estimation between two sensors was reduced to the registration of plane and line. Li [12], Kwak et al. [13] used isosceles triangles and foliate panels as calibration targets, and used point-line constraint to calibrate the camera and LRF. Dong et al. [14] proposed a special V-shaped calibration target method with the checkerboard, which was used for both camera and LRF calibration by a single observation, but it needed a cumbersome solution process. Itami et al. [15,16] proposed an improved method for the checkerboard calibration of the camera and LRF, which directly obtained the point-to-point correspondence between the LRF and camera. Huang et al. [23] proposed a method to calibrate the 2D LRF and camera using a one-side transparent hollow calibration board, but it required the scanning surface of the LRF to form a certain angle with the hollow calibration board. Tu et al. [24] proposed an accuracy criterion for directional synthesis to eliminate the large error data during observation, but their laser and visual observation points were less constrained and still required multiple observations. Although there are many extrinsic parameters calibration methods for the 2D LRF and camera, the problems of a complex calibration process, high requirements for calibration board production, many calibration times and a limited calibration environment still exist.

References [25–31] are the calibration without target, which are further divided into feature-based extrinsic calibration and motion-based extrinsic calibration. Levison et al. [25] proposed a self-calibration method based on edge feature matching. Zhao et al. [26] proposed an extrinsic calibration framework based on motion LRF and visible light camera. Yang et al. [27] built on previous methods to achieve matching of images and 3D LRF point clouds by methods such as keyframing and motion recovery structures. However, since different sensors acquired data on different principles, the transformation of each sensor was determined based on the sensor that acquired data at the lowest frequency. Schneider et al. [28] proposed to apply deep learning to LRF and the extrinsic calibration of the visible light camera, constructing loss functions by photometric loss and point cloud distance loss, and using unsupervised learning methods for training. However, such methods rely on image feature points and radar point cloud data that are often difficult to obtain in natural scenes and have harsh usage conditions. In order to estimate the LiDAR to stereo camera extrinsic parameters for driving platforms, applying 3D mesh reconstruction-based point cloud registration, a photometric error function was built [31]. In addition to directly obtaining the extrinsic parameter of calibration, CFNet proposed by Wang [32] was utilized to predict the calibration flow based on convolutional neural networks. Recently, other authors proposed to optimize the external parameter calibration with additional sensors [33]. The targetless extrinsic parameter calibration has a common feature, where lidar collects 3D point cloud data with rich feature information. So they are too difficult to be applied to the calibration of 2D LRF.

In order to solve the calibration problems mentioned above, in this paper, we propose a method to constrain the correspondence between the depth camera and 2D LRF within a special calibration plate. The point-line features constraint of the 2D LRF and the depth camera on the calibration plate is used to realized the joint calibration of the 2D LRF and depth camera. For the extrinsic calibration of the 2D laser rangefinder and depth camera, the main contributions of the article are as follows:

1. Provide a novel specific calibration board which is simple to manufacture for the 2D LRF and camera calibration to construct three observation feature point-line constraints between two sensors.
2. Through the method in this paper, the joint calibration of 2D LRF and depth camera is completed by only two observations with an oversimplified operation.

3. By setting the calibration threshold, the joint calibration of the 2D LRF and depth camera placed on a movable device adjusts the number of observations autonomously.

The layout of the article is as follows: Section 2 describes the calibration principles of the 2D LRF and depth cameras. Section 3 describes the calibration methods and algorithms for the 2D LRF and depth cameras. Section 4 verifies the proposed method by experiments, and we draw conclusions in Section 5.

## 2. The Calibration Basis of 2D LRF and Depth Camera

In the process of mobile robot localization and mapping, it is often necessary to unify the environmental perception data of each sensor to the world coordinate system called base frame to describe. The joint calibration of 2D LRF and depth camera is to determine the coordinate transformation relationship between the coordinate system of the depth camera, 2D LRF and the world. As shown in Figure 1, the calibration involves four coordinate systems, which are world coordinate system $O_w - x_w y_w z_w$, LRF coordinate system $O_l - x_l y_l z_l$ called laser frame, depth camera coordinate system $O_c - x_c y_c z_c$ called camera frame, and pixel coordinate system $O - uv$. The pixel coordinate system is the reference coordinate system of the camera observation data, which usually needs to be transferred to the camera coordinate system. With the determined relative position among $O_w - x_w y_w z_w$, $O_l - x_l y_l z_l$ and $O_c - x_c y_c z_c$, the observation point coordinates are able to be transformed among the coordinate systems. Depending on the application scenarios and needs, the observation data from the depth camera can be expressed under the LRF coordinate system and then the data can be projected to describe in the world coordinate system. The scanned data of the LRF can be also expressed under the camera coordinate system, and then the data are transformed from the camera coordinate system to the world coordinate system.

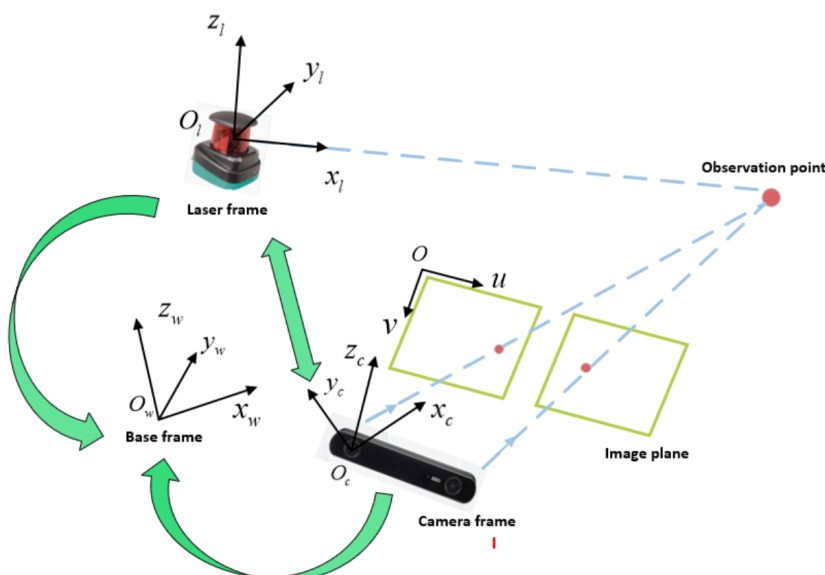

**Figure 1.** Relationship of coordinate systems.

The 2D LRF is often mounted horizontally in mobile robot applications. However, due to installation error and uncertainty, the mounting will have some deflection angles. The camera also has some angles relative to the world coordinate system. According to the rigid body coordinate transformation relationship, the position data collected by the 2D

LRF and depth camera in their respective coordinate systems correspond to the position of the observation point in the world coordinate system, as shown in Equation (1).

$$\begin{bmatrix} x_w \\ y_w \\ z_w \end{bmatrix} = \mathbf{R}_i^w \begin{bmatrix} x_i \\ y_i \\ z_i \end{bmatrix} + \mathbf{T}_i^w, \tag{1}$$

In the formula, $x_i$ is the coordinate value in $i$ coordinate system. $i = l, c, l$ is the radar coordinate system, $c$ is the camera coordinate system. $\mathbf{R}_i^w$ is the rotation matrix from the coordinate system to the world coordinate system, and $\mathbf{T}_i^w$ is the translation matrix of the $i$ coordinate system with respect to the world coordinate system. The camera used in this paper is the ZED depth camera, and its imaging principle is shown in Figure 2. The figure contains two coordinate systems: the camera coordinate system $O_c - x_c y_c z_c$, and the pixel coordinate system $O - uv$. The two cameras of the depth camera are located in the same plane, the optical axes of the left and right cameras are parallel, and the focal length parameters $f$ are the same. The coordinates of the observation point in the camera coordinate system are assumed to be $\mathbf{P}(x_c, y_c, z_c)$.

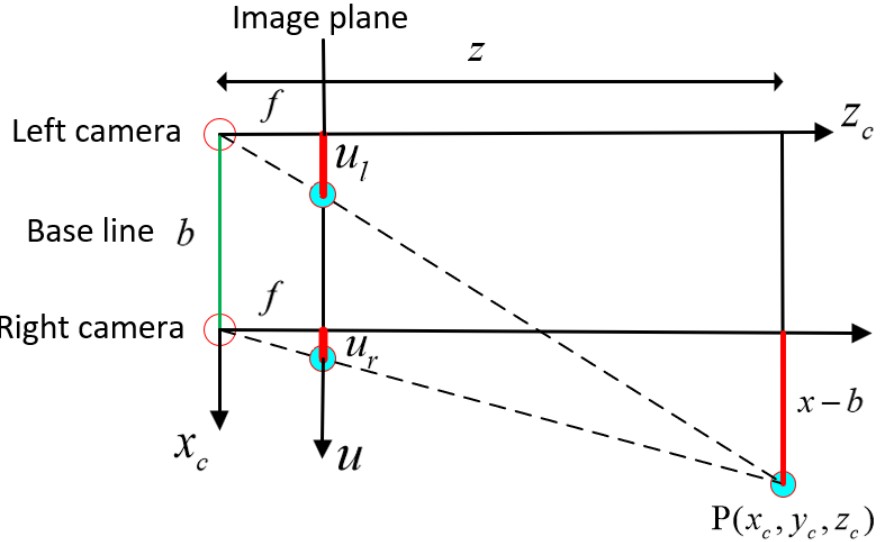

**Figure 2.** Imaging principle of binocular stereo camera.

According to the camera's small-aperture imaging principle and triangle similarity, we have

$$\begin{aligned} \frac{z_c}{f} &= \frac{x_c}{u_l} \\ \frac{z_c}{f} &= \frac{x_c - b}{u_r} \\ \frac{z_c}{f} &= \frac{y_c}{v_l} = \frac{y_c}{v_r} \end{aligned}, \tag{2}$$

where $x_c, y_c, z_c$ are the coordinates of $O_c - x_c y_c z_c$; $u_l$ and $v_l$ are the coordinates in the left camera pixel coordinate system; $u_r$ and $v_r$ are the coordinates in the right camera pixel coordinate system; $f$ is the focal length of the camera; and $b$ isthe distance which is called the base line between the binocular cameras. Using the camera coordinate system of the left camera as the camera coordinate system of the depth camera, the relationship between the pixel coordinates and the camera coordinates of the depth camera is obtained as

$$\begin{aligned} x_c &= \frac{u_l * z_c}{f} \\ y_c &= \frac{v_l * z_c}{f} \\ z_c &= \frac{f * b}{u_l - u_r} \end{aligned}, \tag{3}$$

By Equation (3),

$$\begin{bmatrix} x_c \\ y_c \\ z_c \end{bmatrix} = \begin{bmatrix} \frac{b}{d} & 0 & 0 \\ 0 & \frac{b}{d} & 0 \\ 0 & 0 & \frac{b*f}{d} \end{bmatrix} \begin{bmatrix} u_l \\ v_l \\ 1 \end{bmatrix}, \tag{4}$$

where $d = u_l - u_r$, $d$ is called the parallax of the two cameras. According to the depth data acquisition principle of the depth camera, the point cloud data of the observed object is gathered. For the 2D LRF, it directly obtains the distance and angle information of the obstacle. In practical application, the points of the actual object scanned by the LRF have a unique point corresponding to them in the depth camera. The polar coordinate data of the LRF are converted into Cartesian coordinate data, then the Cartesian coordinates are converted to be expressed under the coordinate system of $O_c - x_c y_c z_c$ by Equation (5).

$$\begin{bmatrix} x_c \\ y_c \\ z_c \end{bmatrix} = \mathbf{R}_l^c \begin{bmatrix} x_l \\ y_l \\ z_l \end{bmatrix} + \mathbf{T}_l^c, \tag{5}$$

where $\mathbf{R}_l^c = \begin{bmatrix} r_{11} & r_{12} & r_{13} \\ r_{21} & r_{22} & r_{23} \\ r_{31} & r_{32} & r_{33} \end{bmatrix}$, $\mathbf{T}_l^c = \begin{bmatrix} t_1 \\ t_2 \\ t_3 \end{bmatrix}$.    $\mathbf{R}_l^c$ denotes the $3 \times 3$ rotation matrix from $O_l - x_l y_l z_l$ to $O_c - x_c y_c z_c$, and $\mathbf{T}_l^c$ denotes the translation matrix from $O_l - x_l y_l z_l$ to $O_c - x_c y_c z_c$.

Because the geometric relationships do not vary with the coordinate system, the coordinate system transformation does not affect the geometric constraint relationships. The data points acquired by the 2D LRF are transformed to be expressed under the depth camera coordinate system, and the equations are constructed using the constraints of the LRF scanned points on the depth camera observation line. The set of equations constructed using multiple observations is solved by linear least squares for the rotation matrix $\mathbf{R}_l^c$ and translation matrix $\mathbf{T}_l^c$.

## 3. Calibration Methods

### 3.1. Feature Extraction

Although the scanned points of 2D LRF are not visible, it accurately obtains the contour of the obstacle. Based on the characteristics of 2D LRF, a specific calibration plate is used in this paper, as shown in Figure 3. The special feature of the calibration plate is its clever shape, which makes the laser scan points form three uncorrelated straight lines and makes the camera's 3D point cloud form three uncorrelated planes. The calibration plate consists of two rectangular planes and two triangular planes. There is no limit to the angle between the various planes in the production of the calibration plate. The only thing that needs to be ensured is that any three planes are uncorrelated and obtain as many observation points as possible on the calibration plate. In addition, the calibration plate eliminates the limitations of the environment and installation relationship during the calibration process, just ensuring that the LRF and camera observe the calibration plate at the same time. By the specific calibration plate, the characteristic information of 2D LRF scanning is obtained. At the same time, the 3D point cloud data of the depth camera observation on calibration plate are obtained too.

As shown in Figure 4, the calibration plate is placed at the position where the 2D LRF and depth camera observe simultaneously, and the observation data of 2D LRF and depth camera are collected. The 2D LRF data are observed based on the LRF coordinate system $O - x_l y_l z_l$, and the observation data of depth camera are observed based on the left camera of depth camera coordinate system $O - x_c y_c z_c$.

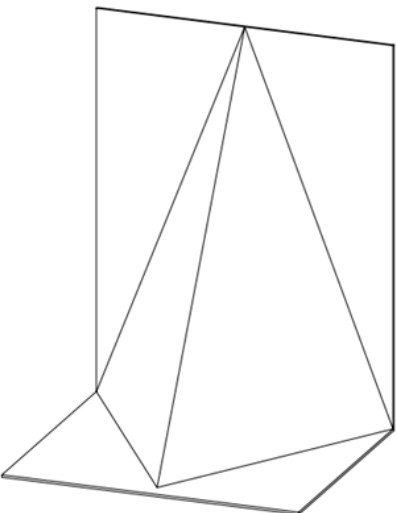

**Figure 3.** Combined 2D LRF and depth camera calibration board.

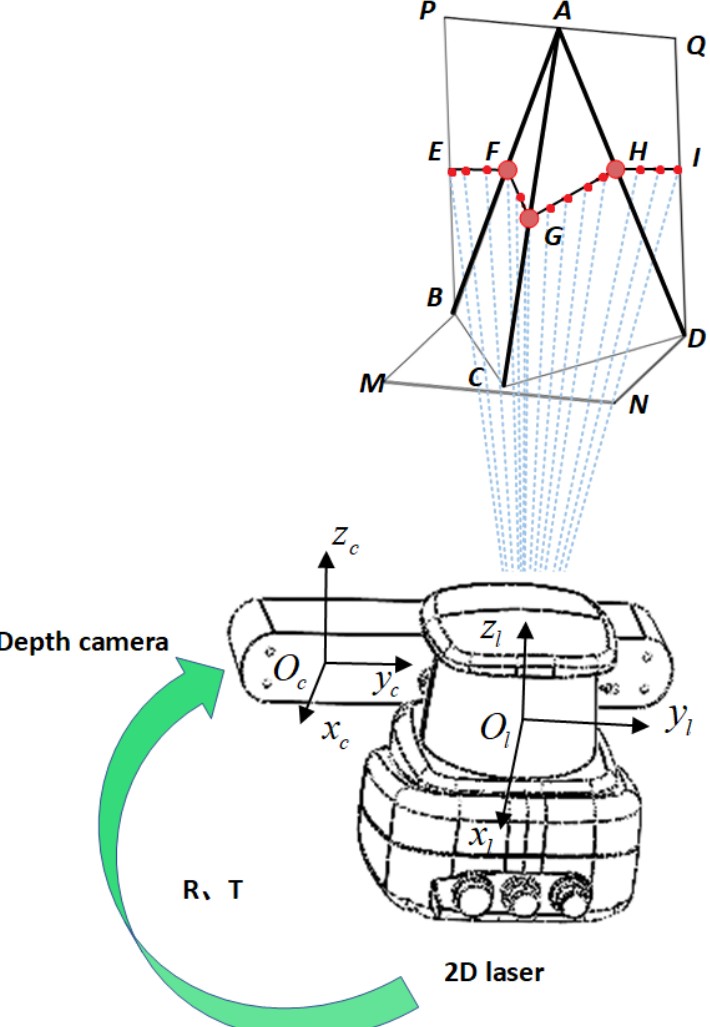

**Figure 4.** Feature point extraction.

On the calibration plate, the scanned points of LRF will form a folded shape of $EFGGHI$, as shown in Figure 4. By the RANSAC (random sample consensus) method, the scanned points on the rectangular plane $PBDQ$ are fitted to the straight line $\overline{EI}$, the points

on the plane $ABC$ are fitted to the straight line $\overline{FG}$, and the points on the plane $ACD$ are fitted to the straight line $\overline{GH}$. Any two of the three lines are respectively associated to find the intersection points $\mathbf{F}(x_1^l, y_1^l, 0)$, $\mathbf{G}(x_2^l, y_2^l, 0)$, $\mathbf{H}(x_3^l, y_3^l, 0)$.

The depth data from the depth camera are transformed into the 3D point cloud, and the point cloud data of plane $PBDQ$, plane $ABC$, and plane $ACD$ under the observation of the depth camera are extracted. By RANSAC (random sample consensus ) method, the 3D point clouds on each plane are fitted to the plane equations of the corresponding planes. The spatial linear equations of straight line $\overline{AB}$, straight line $\overline{AC}$ and straight line $\overline{AD}$ are obtained by associating any two plane equations of plane $PBDQ$, plane $ABC$ and plane $ACD$. Suppose the plane equations of the plane $PBDQ$, plane $ABC$, and plane $ACD$ are

$$
\begin{aligned}
A_1 x^c + B_1 y^c + C_1 z^c + D_1 &= 0 \\
A_2 x^c + B_2 y^c + C_2 z^c + D_2 &= 0 \\
A_3 x^c + B_3 y^c + C_3 z^c + D_3 &= 0
\end{aligned} \quad , \tag{6}
$$

Then the spatial linear equations of the line $\overline{AB}$, line $\overline{AC}$ and line $\overline{AD}$ are

$$
\begin{aligned}
A_1 x^c + B_1 y^c + C_1 z^c + D_1 &= 0 \\
A_2 x^c + B_2 y^c + C_2 z^c + D_2 &= 0 \\
A_2 x^c + B_2 y^c + C_2 z^c + D_2 &= 0 \\
A_3 x^c + B_3 y^c + C_3 z^c + D_3 &= 0 \\
A_3 x^c + B_3 y^c + C_3 z^c + D_3 &= 0 \\
A_1 x^c + B_1 y^c + C_1 z^c + D_1 &= 0
\end{aligned} \quad , \tag{7}
$$

where $A_1, B_1, C_1, A_2, B_2, C_2, A_3, B_3, C_3$ are known. By extracting and fitting 2D LRF data and depth camera point cloud data, the linear equations of three feature points based on the LiDAR coordinate system and three spatial straight lines based on the depth camera coordinate system are obtained.

### 3.2. Parameter Fitting

The projection of the three feature points of 2D LRF into the depth camera coordinate system by Equation (5) has

$$
\begin{bmatrix} x_{li}^c \\ y_{li}^c \\ z_{li}^c \end{bmatrix} = \begin{bmatrix} r_{11} x_i^l + r_{12} y_i^l + t_1 \\ r_{21} x_i^l + r_{22} y_i^l + t_2 \\ r_{31} x_i^l + r_{32} y_i^l + t_3 \end{bmatrix}, \tag{8}
$$

where $l$ denotes the 2D LRF coordinate system, $i$ denotes the i-th intersection point under one 2D LRF observation, and $i = 1, 2, 3$. From the projection results, it is known that only nine unknowns need to be solved to obtain the rotation and translation matrices for the 2D LRF and depth camera coordinate system conversions, so only nine mutually independent sets of equations need to be coupled. Since the transformation of the coordinate system of points does not change the geometric relationship, the points under the 2D LRF coordinate system should be on the straight line $\overline{AB}$, the straight line $\overline{AC}$ and the straight line $\overline{AD}$, respectively, after the points are transformed to the camera coordinate system, i.e.,

$$
\begin{aligned}
A_1 r_{11} x_1^l + A_1 r_{12} y_1^l + A_1 t_1 + B_1 r_{21} x_1^l + B_1 r_{22} y_1^l + B_1 t_2 + C_1 r_{31} x_1^l + C_1 r_{32} y_1^l + C_1 t_3 + D_1 &= 0 \\
A_2 r_{11} x_1^l + A_2 r_{12} y_1^l + A_2 t_1 + B_2 r_{21} x_1^l + B_2 r_{22} y_1^l + B_2 t_2 + C_2 r_{31} x_1^l + C_2 r_{32} y_1^l + C_2 t_3 + D_2 &= 0 \\
A_2 r_{11} x_2^l + A_2 r_{12} y_2^l + A_2 t_1 + B_2 r_{21} x_2^l + B_2 r_{22} y_2^l + B_2 t_2 + C_2 r_{31} x_2^l + C_2 r_{32} y_2^l + C_2 t_3 + D_2 &= 0 \\
A_3 r_{11} x_2^l + A_3 r_{12} y_2^l + A_3 t_1 + B_3 r_{21} x_2^l + B_3 r_{22} y_2^l + B_3 t_2 + C_3 r_{31} x_2^l + C_3 r_{32} y_2^l + C_3 t_3 + D_3 &= 0 \\
A_3 r_{11} x_3^l + A_3 r_{12} y_3^l + A_3 t_1 + B_3 r_{21} x_3^l + B_3 r_{22} y_3^l + B_3 t_2 + C_3 r_{31} x_3^l + C_3 r_{32} y_3^l + C_3 t_3 + D_3 &= 0 \\
A_1 r_{11} x_3^l + A_1 r_{12} y_3^l + A_1 t_1 + B_1 r_{21} x_3^l + B_1 r_{22} y_3^l + B_1 t_2 + C_1 r_{31} x_3^l + C_1 r_{32} y_3^l + C_1 t_3 + D_1 &= 0
\end{aligned} \quad , \tag{9}
$$

Collated by

$$
\begin{bmatrix}
A_1 x_1^l & A_1 y_1^l & A_1 & B_1 x_1^l & B_1 y_1^l & B_1 & C_1 x_1^l & C_1 y_1^l & C_1 \\
A_2 x_1^l & A_2 y_1^l & A_2 & B_2 x_1^l & B_2 y_1^l & B_2 & C_2 x_1^l & C_2 y_1^l & C_2 \\
A_2 x_2^l & A_2 y_2^l & A_2 & B_2 x_2^l & B_2 y_2^l & B_2 & C_2 x_2^l & C_2 y_2^l & C_2 \\
A_3 x_2^l & A_3 y_2^l & A_3 & B_3 x_2^l & B_3 y_2^l & B_3 & C_3 x_2^l & C_3 y_2^l & C_3 \\
A_3 x_3^l & A_3 y_3^l & A_3 & B_3 x_3^l & B_3 y_3^l & B_3 & C_3 x_3^l & C_3 y_3^l & C_3 \\
A_1 x_3^l & A_1 y_3^l & A_1 & B_1 x_3^l & B_1 y_3^l & B_1 & C_1 x_3^l & C_1 y_3^l & C_1
\end{bmatrix}
\begin{bmatrix}
r_{11} \\ r_{12} \\ t_1 \\ r_{21} \\ r_{22} \\ t_2 \\ r_{31} \\ r_{32} \\ t_3
\end{bmatrix}
=
\begin{bmatrix}
-D_1 \\ -D_2 \\ -D_2 \\ -D_3 \\ -D_3 \\ -D_1
\end{bmatrix}, \qquad (10)
$$

The six equations in Equation (9) are independent of each other. By changing the calibration model or the position and attitude of the calibration plate, the six constraint equations are obtained again by the same procedure. Since there are only nine unknowns in the calibration parameters, the set of equations from the last observation is combined to form the super-stationary set of equations in Equation (11), where $n$ is the number of observations. The parameters of the rotation and translation matrices are solved using linear least squares to determine the coordinate transformation of the 2D LRF and depth camera.

$$
\begin{bmatrix}
A_{11} x_{11}^l & A_{11} y_{11}^l & A_{11} & B_{11} x_{11}^l & B_{11} y_{11}^l & B_{11} & C_{11} x_{11}^l & C_{11} y_{11}^l & C_{11} \\
A_{12} x_{11}^l & A_{12} y_{11}^l & A_{12} & B_{12} x_{11}^l & B_{12} y_{11}^l & B_{12} & C_{12} x_{11}^l & C_{12} y_{11}^l & C_{12} \\
A_{12} x_{12}^l & A_{12} y_{12}^l & A_{12} & B_{12} x_{12}^l & B_{12} y_{12}^l & B_{12} & C_{12} x_{12}^l & C_{12} y_{12}^l & C_{12} \\
A_{13} x_{12}^l & A_{13} y_{12}^l & A_{13} & B_{13} x_{12}^l & B_{13} y_{12}^l & B_{13} & C_{13} x_{12}^l & C_{13} y_{12}^l & C_{13} \\
A_{13} x_{13}^l & A_{13} y_{13}^l & A_{13} & B_{13} x_{13}^l & B_{13} y_{13}^l & B_{13} & C_{13} x_{13}^l & C_{13} y_{13}^l & C_{13} \\
A_{11} x_{13}^l & A_{11} y_{13}^l & A_{11} & B_{11} x_{13}^l & B_{11} y_{13}^l & B_{11} & C_{11} x_{13}^l & C_{11} y_{13}^l & C_{11} \\
\vdots & \vdots & \vdots & \vdots & \vdots & \vdots & \vdots & \vdots & \vdots \\
A_{n1} x_{n1}^l & A_{n1} y_{n1}^l & A_{n1} & B_{n1} x_{n1}^l & B_{n1} y_{n1}^l & B_{n1} & C_{n1} x_{n1}^l & C_{n1} y_{n1}^l & C_{n1} \\
A_{n2} x_{n1}^l & A_{n2} y_{n1}^l & A_{n2} & B_{n2} x_{n1}^l & B_{n2} y_{n1}^l & B_{n2} & C_{n2} x_{n1}^l & C_{n2} y_{n1}^l & C_{n2} \\
A_{n2} x_{n2}^l & A_{n2} y_{n2}^l & A_{n2} & B_{n2} x_{n2}^l & B_{n2} y_{n2}^l & B_{n2} & C_{n2} x_{n2}^l & C_{n2} y_{n2}^l & C_{n2} \\
A_{n3} x_{n2}^l & A_{n3} y_{n2}^l & A_{n3} & B_{n3} x_{n2}^l & B_{n3} y_{n2}^l & B_{n3} & C_{n3} x_{n2}^l & C_{n3} y_{n2}^l & C_{n3} \\
A_{n3} x_{n3}^l & A_{n3} y_{n3}^l & A_{n3} & B_{n3} x_{n3}^l & B_{n3} y_{n3}^l & B_{n3} & C_{n3} x_{n3}^l & C_{n3} y_{n3}^l & C_{n3} \\
A_{n1} x_{n3}^l & A_{n1} y_{n3}^l & A_{n1} & B_{n1} x_{n3}^l & B_{n1} y_{n3}^l & B_{n1} & C_{n1} x_{n3}^l & C_{n1} y_{n3}^l & C_{n1}
\end{bmatrix}_{6n \times 9}
\begin{bmatrix}
r_{11} \\ r_{12} \\ t_1 \\ r_{21} \\ r_{22} \\ t_2 \\ r_{31} \\ r_{32} \\ t_3
\end{bmatrix}_{9 \times 1}
=
\begin{bmatrix}
-D_{11} \\ -D_{12} \\ -D_{12} \\ -D_{13} \\ -D_{13} \\ -D_{11} \\ \vdots \\ -D_{n1} \\ -D_{n2} \\ -D_{n2} \\ -D_{n3} \\ -D_{n3} \\ -D_{n1}
\end{bmatrix}_{6n \times 1}, \quad (11)
$$

### 3.3. Calibration Algorithm

For the joint calibration of the 2D LRF and depth camera, the method in this paper aims to determine the coordinate transformation relationship between the depth camera, LRF and the world. Before the calibration work, the calibrated depth camera had completed internal parameter calibration. To evaluate the calibration accuracy of the method in this paper, the three feature points under the LRF observation are projected under the point cloud reference coordinate system of the depth camera by solving the obtained calibration parameters, and the projection error of the laser points is calculated using the distance from the formula point to the straight line. The calibration plate or calibration model's position is changed, the experiment is repeated several times, and the average calibration accuracy is calculated by multiple sets of data.

$$
err = \frac{1}{3N} \sum_{i=1}^{N} \sum_{j=1}^{3} \frac{|A_j x_j + B_j y_j + C_j z_j + D_j|}{\sqrt{A_j^2 + B_j^2 + C_j^2}}, \qquad (12)
$$

where $N$ is the number of test, $(x_j, y_j, z_j)$ is the coordinate of the point projected to the depth camera coordinate system. $A_j, B_j, C_j, D_j$ is the linear equation coefficient of the corresponding line of the projected point. $err$ is the calibration accuracy. The smaller the $err$ is, the higher the calibration accuracy.

The algorithm framework of the calibration method in this paper is shown in Algorithm 1, where **R**, **T** and $ObservNum$ are defined as the final extrinsic parameter calibration result.

Datasets *I* and *S* are the point cloud of the depth camera and 2D LRF. Dataset *P* is the pose sequence of AGV. Additionally, $\mathcal{E}$ is the error threshold, and $\mathbf{R}gt$ and $\mathbf{T}gt$ are the ground truth of the rotation and translation matrices. After two observations, the algorithm first solves the calibration error and compares it with the calibration threshold. When the calibration error is less than the target threshold, the rotation matrix $\mathbf{R}$, the translation matrix $\mathbf{T}$ and the number of observations are output to complete the extrinsic parameter calibration of LRF and depth camera.

---

**Algorithm 1** LRF and depth camera extrinsic parameter calibration.

---

**Input:** Image set *I*, point cloud set *S*, calibration position of AGV set *P*, and error threshold $\mathcal{E}$, ground truth $\mathbf{R}gt$, ground truth $\mathbf{T}gt$

**Output:** Calibration result $\mathbf{R}$, $\mathbf{T}$ and *Observ*

1: $\mathbf{R} \leftarrow 0, \mathbf{T} \leftarrow 0, Observ \leftarrow 0$

2: $\mathbf{A} \leftarrow 0, \mathbf{b} \leftarrow 0, err \leftarrow 100$

3: $AngularError \leftarrow 0$

4: $DistanceError \leftarrow 0$

5: $\mathbf{X} \leftarrow \begin{bmatrix} r_{11} & r_{12} & t_1 & r_{21} & r_{22} & t_2 & r_{31} & r_{32} & t_3 \end{bmatrix}^{\text{T}}$

6: Set firt calibration position of AGV from *P*

7: $Observ \leftarrow 1$

8: **while** $err > \mathcal{E}$ **do**

9:     Get set *I* and *S*

10:     Extracted 3D point cloud of calibration plate features

11:     Calculate $\mathbf{M} \leftarrow \begin{bmatrix} A_1 & B_1 & C_1 & D_1 \\ A_2 & B_2 & C_2 & D_2 \\ A_3 & B_3 & C_3 & D_3 \end{bmatrix}$

12:     Extracted 2D point cloud of calibration plate shape features

13:     Calculate $\mathbf{L} \leftarrow \begin{bmatrix} x_1^l & y_1^l \\ x_2^l & y_2^l \\ x_3^l & y_3^l \end{bmatrix}$

14:     Calculate $\mathbf{C} \leftarrow \begin{bmatrix} x_1^c & y_1^c \\ x_2^c & y_2^c \\ x_3^c & y_3^c \end{bmatrix}$

15:     $A1 \leftarrow \begin{bmatrix} A_1x_1^l & A_1y_1^l & A_1 & B_1x_1^l & B_1y_1^l & B_1 & C_1x_1^l & C_1y_1^l & C_1 \\ A_2x_1^l & A_2y_1^l & A_2 & B_2x_1^l & B_2y_1^l & B_2 & C_2x_1^l & C_2y_1^l & C_2 \\ A_2x_2^l & A_2y_2^l & A_2 & B_2x_2^l & B_2y_2^l & B_2 & C_2x_2^l & C_2y_2^l & C_2 \\ A_3x_2^l & A_3y_2^l & A_3 & B_3x_2^l & B_3y_2^l & B_3 & C_3x_2^l & C_3y_2^l & C_3 \\ A_3x_3^l & A_3y_3^l & A_3 & B_3x_3^l & B_3y_3^l & B_3 & C_3x_3^l & C_3y_3^l & C_3 \\ A_1x_3^l & A_1y_3^l & A_1 & B_1x_3^l & B_1y_3^l & B_1 & C_1x_3^l & C_1y_3^l & C_1 \end{bmatrix}$

16:     $\mathbf{b1} \leftarrow \begin{bmatrix} -D_1 & -D_2 & -D_2 & -D_3 & -D_3 & -D_1 \end{bmatrix}^{\text{T}}$

17:     **if** *Observ*>2 **then**

18:         $err \leftarrow \frac{1}{3} \sum_{j=1}^{3} \frac{\left| A_jx_j^c + B_jy_j^c + C_jz_j^c + D_j \right|}{\sqrt{A_j^2 + B_j^2 + C_j^2}}$

19:         **if** $err < \mathcal{E}$ **then**

20:             Calculate $\mathbf{R} \leftarrow \begin{bmatrix} r_{11} & r_{12} & r_{13} \\ r_{21} & r_{22} & r_{23} \\ r_{31} & r_{32} & r_{33} \end{bmatrix}$

21:             Calculate $\mathbf{T} \leftarrow \begin{bmatrix} t_1 & t_2 & t_3 \end{bmatrix}^{\text{T}}$

22:             $AngularError \leftarrow \cos^{-1}\left( \frac{\text{trace}\left( \mathbf{R}_{gt}^{-1}\mathbf{R} \right) - 1}{2} \right)$

---

**Algorithm 1** *Cont.*

| | |
|---|---|
| 23: | $DistanceError \leftarrow \left\| \mathbf{T} - \mathbf{T}_{gt} \right\|_2$ |
| 24: | **else** |
| 25: | Add **A**1 into **A** |
| 26: | Add **b**1 into **b** |
| 27: | Calculate **X** by $\mathbf{AX} = \mathbf{b}$ |
| 28: | Set another calibration of AGV position from **P** |
| 29: | $Observ \leftarrow Observ + 1$ |
| 30: | **end if** |
| 31: | **else** |
| 32: | Add **A**1 into **A** |
| 33: | Add **b**1 into **b** |
| 34: | **if** $Observ = 2$ **then** |
| 35: | Calculate **X** by $\mathbf{AX} = \mathbf{b}$ |
| 36: | **end if** |
| 37: | Set another calibration board position from **P** |
| 38: | $Observ \leftarrow Observ + 1$ |
| 39: | **end if** |
| 40: | **end while** |
| 41: | **return R**, **T** and *Observ* |

## 4. Calibration Experiments and Analysis of Results

### 4.1. Experimental Equipment and Environment

The experiments in this paper are based on a ROS (robot operating system) under the Linux environment. As shown in Figure 5a, Pepperl+Fuchs R2000 2D LRF and a stereolabs ZED2i stereo binocular camera are used to collect the 2D point cloud data from the LRF and the 3D point cloud data from the depth camera, respectively. The detailed parameters of depth camera and LRF are shown in Tables 1 and 2. In Experiment, both the depth camera and LRF select a 30 Hz sampling rate.

**Table 1.** 2D LRF basic parameters.

| Range/m | Rate/Hz | Resolution/° | Accuracy/mm | Angle/° |
|---------|---------|--------------|-------------|---------|
| 0.1–30 | 10–50 | 0.042 | ±25 | 360 |

**Table 2.** Depth camera basic parameters.

| Depth Range/m | Depth FPS/Hz | Resolution | Aperture | Field/° |
|---------------|--------------|------------|----------|---------|
| 0.2–20 | 15–100 | 3840 × 1080 | f/1.8 | 110H × 70V × 120D |

The calibration experiments are performed on a homemade automated guided vehicles (AGV) vehicle. The installation relationship of the experimental equipment is shown in Figure 5. The 2D LRF is installed behind and above the depth camera. It is basically installed horizontally. The camera is installed horizontally in front of the 2D LRF, and the relative position of the depth camera and the LRF is kept constant during the experiment. The calibration plate is placed at the location where the 2D LRF and the depth camera observe simultaneously. Through the calibration algorithm proposed in this paper, experimental data under different observation positions, which are adjusted by controlling the AGV, are collected to complete the autonomous joint calibration of the 2D LRF and the depth camera.

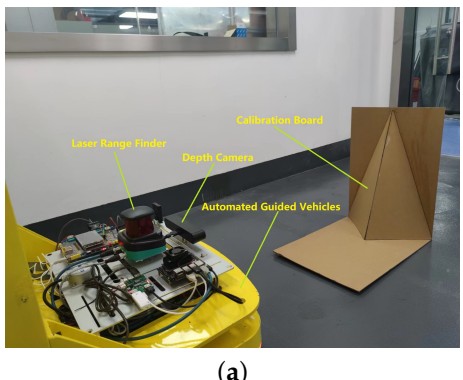 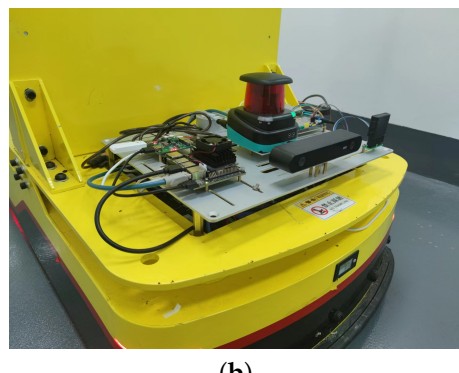

(**a**)                                      (**b**)

**Figure 5.** Experimental equipment and sensor installation location. (**a**) Experimental equipment. (**b**) Relationship of sensor mounting position.

### 4.2. Experimental Steps

The experiments in this paper are performed in a ROS environment. After running the function package of 2D LRF and depth camera, a node and two topics are established to receive respectively the scan topic of 2D LRF and the depth image topic of the depth camera. Then the depth data of the depth camera are converted into 3D point cloud data. The calibration of the 2D LRF and depth camera are completed by executing the following steps in the calibration program.

1. Identify and extract the point cloud data gathered by 2D LRF on the calibration plate by line and corner feature detection algorithms; split the point cloud data into three parts; fit the point cloud of each part into a straight line; and solve the intersection point of any two straight lines. The feature extraction process is shown in Figure 6.
2. Project the intersection points found in the previous step under the depth camera coordinate system by Equation (8).
3. Identify and extract the point cloud collected by the depth camera on the calibration plate by edge and corner detection algorithms. Segment the three planes of the calibration plate; obtain the equation of the plane by fitting the point cloud on the plane; and find the equation of the intersection line between two planes in the three planes. The feature extraction and fitting process are shown in Figure 7.
4. Using the point on the line as a constraint, the coordinates of the projected point are substituted into the intersection equation to obtain six equations.
5. Move the AGV to adjust the observation position, and complete the data collection and extraction again.
6. Solve the rotation and translation matrices of the depth camera and 2D LRF coordinate transformation according to Equation (11).
7. Take multiple experiments, calculate the average of the calibration results.

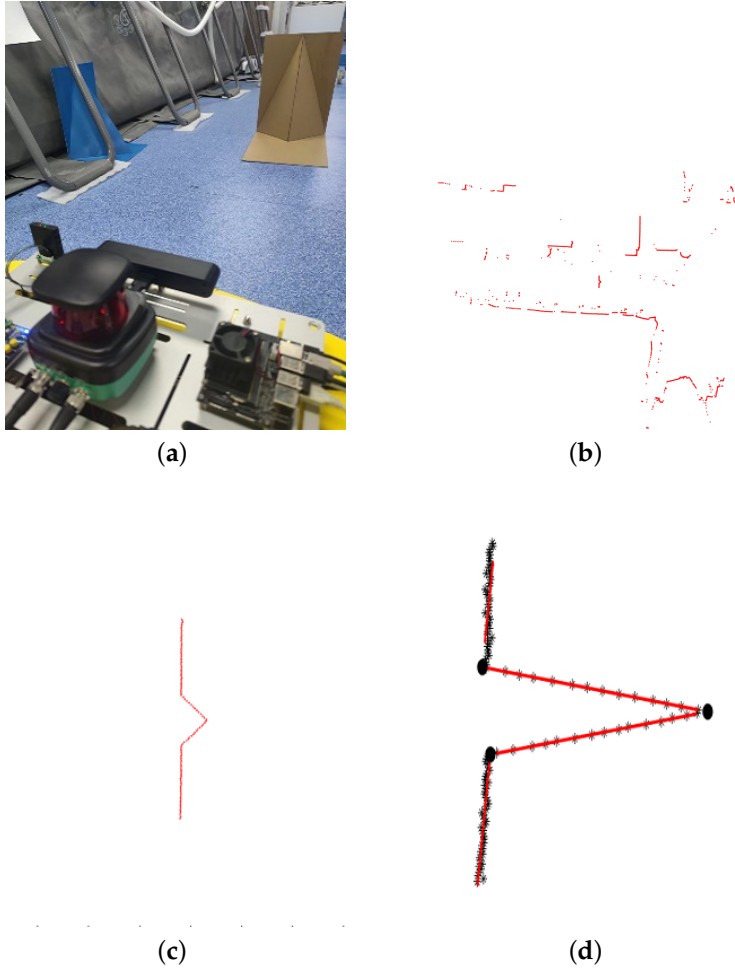

**Figure 6.** Features of LRF extraction and fitting. (**a**) Laser observation position. (**b**) Laser observation data. (**c**) 2D point cloud of calibration plate shape features. (**d**) Feature data of calibration plate fitting.

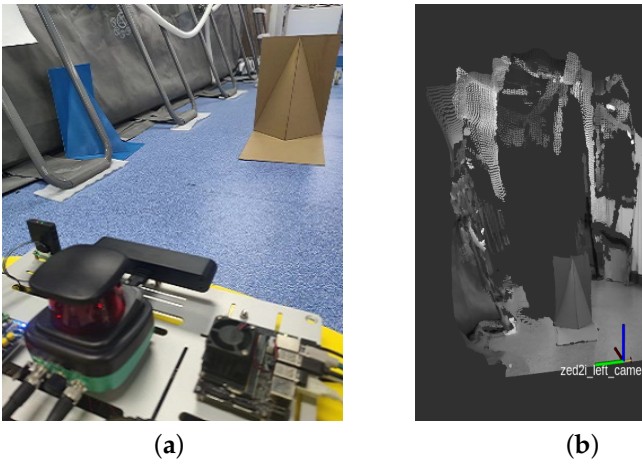

**Figure 7.** *Cont.*

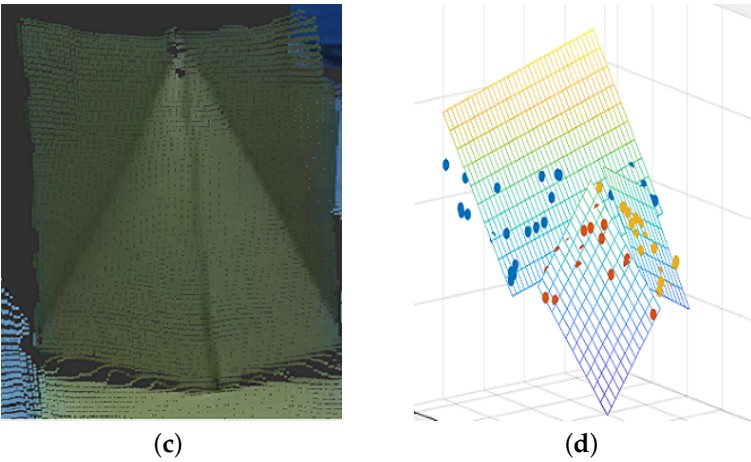

**Figure 7.** Features of depth camera extraction and fitting. (**a**) Depth camera observation position. (**b**) Depth camera point cloud. (**c**) 3D point cloud of calibration plate features. (**d**) 3D point cloud plane fitting of calibration plate.

### 4.3. Experimental Results and Analysis

In this paper, the baseline of ZED2i camera is used as the ground truth. The LRF is first calibrated with the left camera frame to obtain $\mathbf{R}_L^{C_l}$ and $\mathbf{T}_L^{C_l}$, and calibrated with right camera frame to obtain $\mathbf{R}_L^{C_r}$ and $\mathbf{T}_L^{C_r}$. We then compute the relative pose (baseline) between the binocular cameras and compare it with the ground truth $\mathbf{R}_{C_r}^{C_l} = \begin{bmatrix} 1 & 0 & 0 \\ 0 & 1 & 0 \\ 0 & 0 & 1 \end{bmatrix}$ and $\mathbf{T}_{C_r}^{C_l}$

$= \begin{bmatrix} 0 & 120 & 0 \end{bmatrix}^T$ from ZED2i Camera parameters. In order to verify the calibration accuracy and calibration efficiency of the calibration method proposed in this paper, the number of observation is used as the independent variable. For each independent variable, 10 repeated experiments were conducted to obtain the mean values of rotation and translation errors and standard deviation. Based on the experiments, we obtain the relationship between the result of the mean values and standard deviation, and the number of observations as shown in Figure 8.

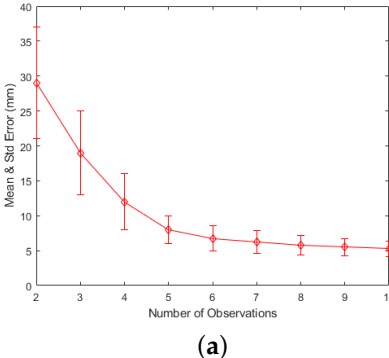

(**a**)

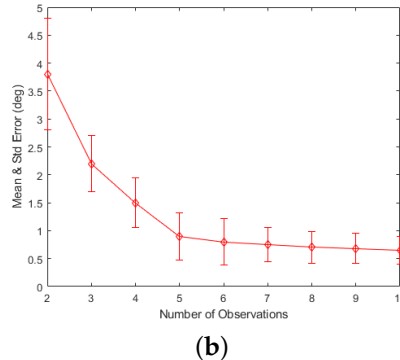

(**b**)

**Figure 8.** Rotation and translation errors versus number of observations. (**a**) Mean and standard deviation of translational errors. (**b**) Mean and standard deviation of rotation errors.

As in Figure 8, the mean values of the rotation and translation errors and the standard deviations of the calibration results gradually decrease as the number of observations increases. After the number of observations is greater than 6, the average value of the rotation error is less than 1°. With the increasing number of observations, the average value of the error gradually tends to be flat. The mean value of the translation error is obtained as 3.74° of the rotation error and 28.31 mm of translation error under only two

observation experiments. The experimental data show that the calibration method of this paper is feasible and achieves high calibration accuracy.

In order to verify the feasibility of this paper's method, the calibration accuracy of our method is compared with the more representative methods of current 2D LRF and camera calibration work. For the methods of Refs. [13] and [23] and the method of this paper, each method divides the experiments into three groups of 2 observations, 10 observations, and 20 observations, and each group of experiments is conducted 10 times. For each method, the LRF is first calibrated for both left and right cameras to obtain the corresponding rotation and translation matrix. Then, the coordinate transformation matrix between the two cameras is calculated and compared with the ground truths $\mathbf{R}_{C_r}^{C_l}$ and $\mathbf{T}_{C_r}^{C_l}$. The average of the 10 experimental results is used as the calibration accuracy. The calibration comparison results are shown in Figure 9. From the experimental results, it is seen that under only two observations, the method in this paper obtains smaller rotation and translation errors than the other two methods. Under multiple observation experiments, the method in this paper achieves the average value of rotation error of 0.68° and the average value of translation error of 6.67 mm, which is better than the other two methods. Finally, the calibration results are shown in Table 3, where $_{C_r}^{C_l}\mathbf{P}_{gt}$ is the ground truth of the relative position of the left and right cameras, and $_{C_r}^{C_l}\mathbf{P}_{calib}$ is the calibration result of the relative position of the left and right cameras. $_{L}^{C_l}\mathbf{P}_{des}$ is the designed installation position of the LRF and depth camera, and $_{L}^{C_l}\mathbf{P}_{calib}$ is the calibration result of the relative position of the LRF and camera.

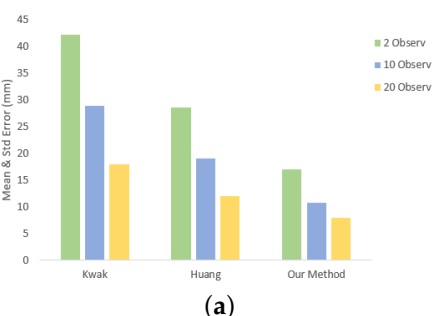
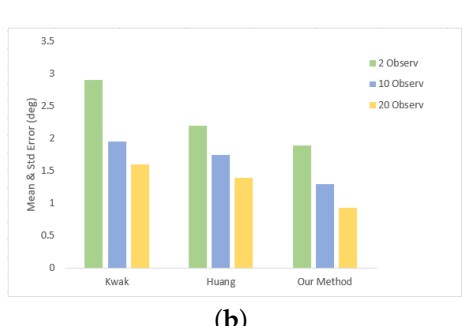

**(a)**           **(b)**

**Figure 9.** Comparison with the methods of Kwak and Huang. (**a**) Mean of translational errors. (**b**) Mean of rotation errors.

**Table 3.** Calibration results.

| Pose | X/mm | Y/mm | Z/mm | Yaw/° | Pitch/° | Roll/° |
|---|---|---|---|---|---|---|
| $_{C_r}^{C_l}\mathbf{P}_{gt}$ | 0 | 120 | 0 | 0 | 0 | 0 |
| $_{C_r}^{C_l}\mathbf{P}_{calib}$ | 1.27 | 125.75 | 3.13 | 0 | 0 | 0.11 |
| $_{L}^{C_l}\mathbf{P}_{des}$ | 128 | 60 | 10 | 0 | 0 | 0 |
| $_{L}^{C_l}\mathbf{P}_{calib}$ | 162.35 | 72.51 | 32.58 | 0 | 0 | 0.57 |

Based on the calibration results, the LRF data are projected into the depth color point cloud map of the depth camera, and the effects before and after calibration are compared visually. The effect before calibration is shown in Figure 10a,b. Before calibration, the 2D lidar point cloud is obscured by the depth point cloud of the camera, and the position of the laser point cloud is lower than the actual observation position. The effect after calibration is shown in Figure 10c,d. After calibration, the point cloud data of the depth camera and the 2D LRF point cloud data basically overlap in the given dimension, which basically meets the application requirements. Due to the occlusion of the vehicle body model, the tf coordinates of the reference system in ROS are shown additionally in Figure 11.

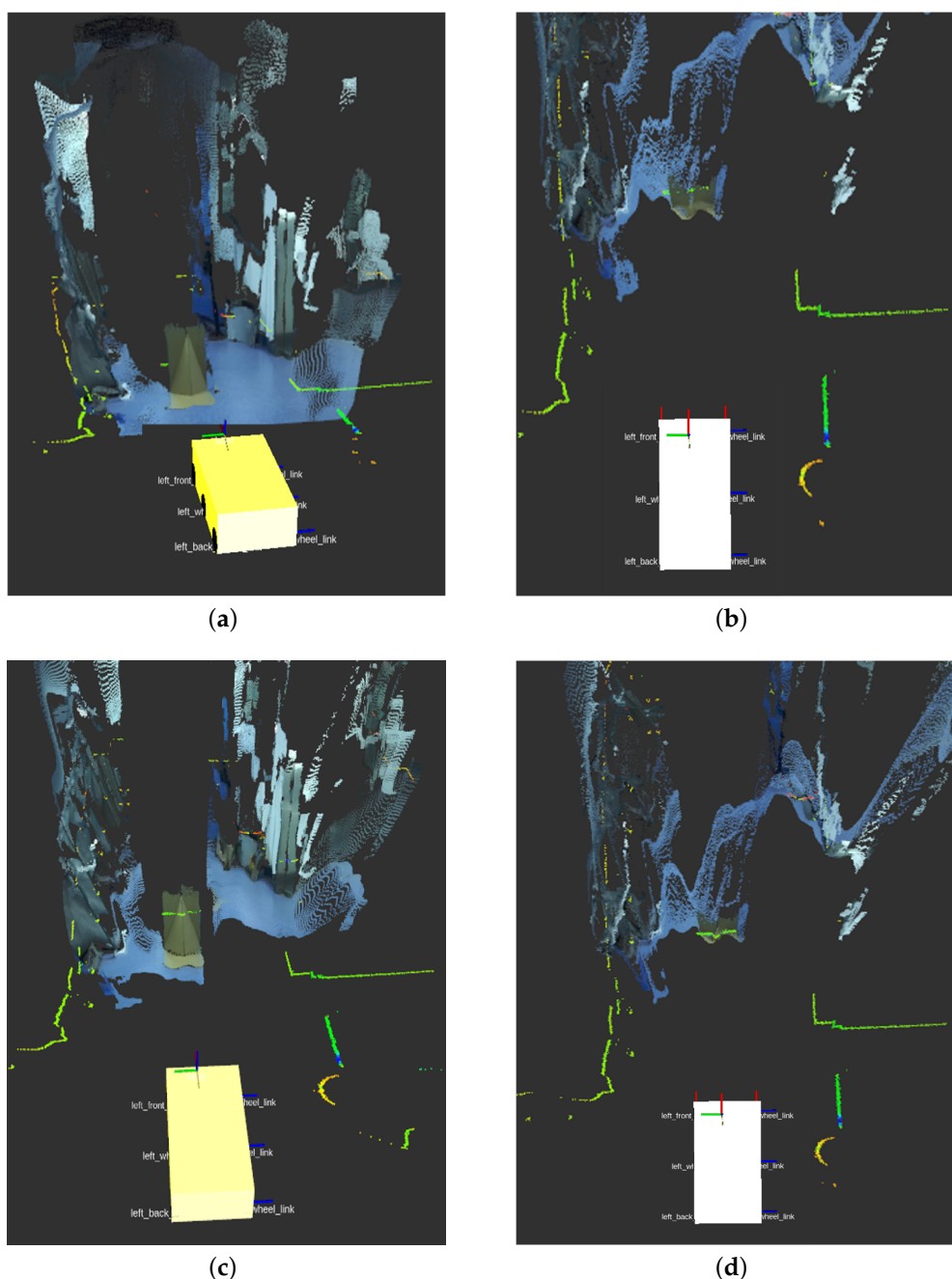

**Figure 10.** Calibration effect of depth camera and 2D LRF. (**a**) Front view of before calibration. (**b**) Top view before calibration. (**c**) Front view after calibration. (**d**) Top view after calibration.

The calibration method in this paper is experimentally verified to be simple and easy to implement, and the joint calibration of the 2D LRF and depth camera is completed with high accuracy and efficiency without excessive position and angle observations. Compared with Ref. [23], there is no complicated feature point extraction and display operation in the calibration process, and no need to adjust the calibration plate position manually. The algorithm is designed to adaptively adjust the AGV observation poses. The calibration method has no restriction on the relative installation position of 2D LRF and depth camera, and the calibration plate is easy and convenient to make. It greatly simplifies the calibration process of the 2D LRF and depth camera and improves the calibration efficiency. Compared with Refs. [13] and [23], the calibration method in this paper has obvious advantages in calibration accuracy, efficiency and operability.

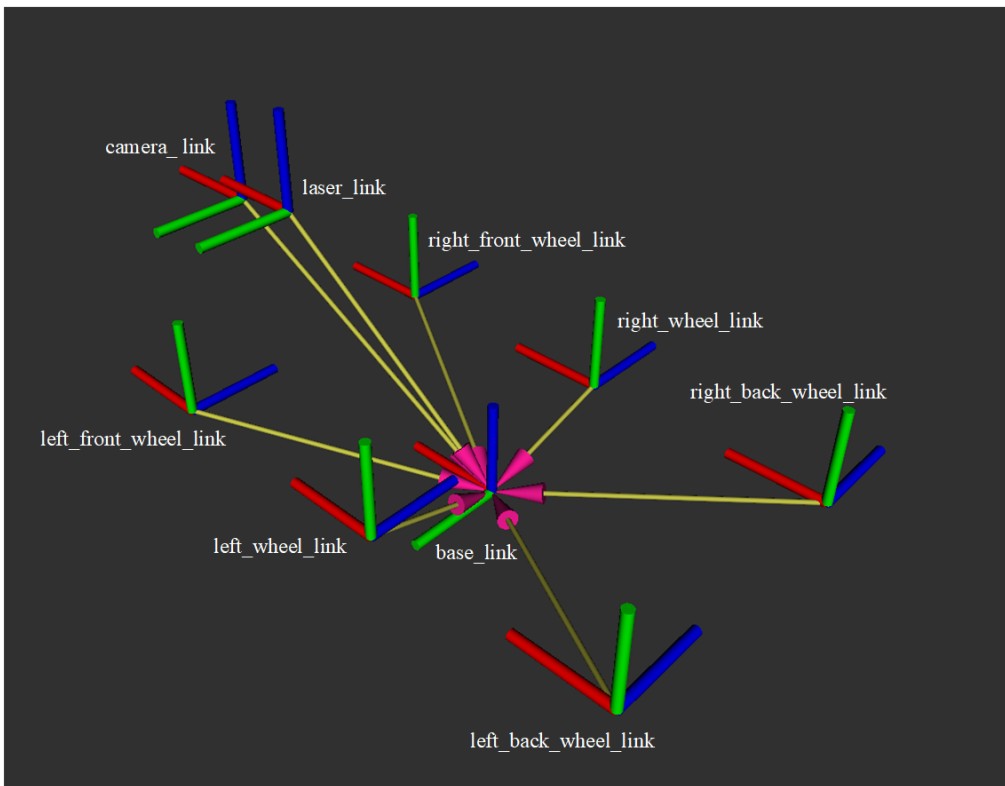

**Figure 11.** Tf coordinate relationship of reference system in ROS.

## 5. Conclusions

In this paper, we present a novel and simplified calibration method to construct point-line constraints for 2D LRF and depth camera observation data by using a specific calibration plate, which effectively achieves the fusion of 2D LRF point cloud data and depth camera 3D point cloud data. The specific calibration plate proposed in this paper eliminates the influence of environment, calibration plate fabrication limitation and sensor installation limitation on calibration. Compared with previous methods, we greatly simplify the calibration process of depth camera and 2D LRF by automatically adjusting the number of observations with a defined error threshold. A series of experiments verify that our method achieves higher accuracy than the compared methods in two observations. Our method is also extended to the case of multiple observations to reduce noise. The calibration accuracy, efficiency and operability of the calibration method meet the practical requirements of mobile robots.

**Author Contributions:** W.Z. and K.H. contributed to the main idea of this paper; H.C. and Z.J. prepared the experimental platform; W.Z. performed the experiments, analyzed the data and wrote the paper. K.H. is the project principal investigator; Q.Z. and Y.X. gave some data analysis and revision suggestions. All authors have read and agreed to the published version of the manuscript.

**Funding:** This research was funded by NSFC-Shenzhen Robot Basic Research Center project (U2013204) and SIAT-CUHK Joint Laboratory of Precision Engineering.

**Institutional Review Board Statement:** Not applicable.

**Informed Consent Statement:** Not applicable.

**Data Availability Statement:** Not applicable.

**Conflicts of Interest:** The authors declare no conflict of interest.

## Abbreviations

The following abbreviations are used in this manuscript:

| | |
|---|---|
| LRF | Laser Range Finder |
| 2D | Two-Dimensional |
| LSQ | Least Square |
| P3P | Perspective Three-Point |
| AGV | Automated Guided Vehicles |

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
