# Peer review of "A Novel and Simplified Extrinsic Calibration of 2D Laser Rangefinder and Depth Camera"

_machines, doi:10.3390/machines10080646_

Round 1

Reviewer 1 Report

This paper is technically correct. Major revision is required.

Abstract: The issue has to be explained better.

Figure 1 has to be explained using more details and more clear reference system. The overall explainationof the picture is not clear.

Punctuation in the formulae is absent.

The mathematical formalism must be adjusted. Scalars cannot be distinguished from vectors and matrices.
Also enter the domain to which the mathematical parameters belong.

Figure 2 is not clear.

Figure 11 (a,b,c, and d), the reference system is absent.

Author Response

Thank you very much for your comments and suggestions, please refer to the attachment for specific responses.

Reviewer 2 Report

The manuscript is well written and clearly describes an approach for extrinsic calibration of 2D laser rangefinder and depth camera. However, the current manuscript requires revisions. 

The main comments are given as follows: 

1) The authors have mentioned a specific calibration plate is used in this paper, as shown in Figure 3. Did the authors design this calibration plate on their own? If yes, please provide more details of this design, such as the motivation and the design specification. If this design is from other literature, please provide the reference. 

2) The writeup of section 3.3 needs major reworking. As far as the reviewer has known it, the camera calibration process is typically a numerical optimization problem, where the objective function (usually can refer to loss function), optimization process (e.g., minimize/maximize the objective), and how this optimization works (e.g., iteration and what the criteria is as model convergence, and etc.? ) should be clearly describled in mathematical form. If there is a novel approach developed in this study, please generalize it mathematically as well. The current writeup does not include any of this information. Please revise. 

3) Same question arises to Figure 5. The flowchart seems more likely to be seen in user manual of a product instead of in a scientific paper...

Considering the flowchart itself, the reviewer understands the authors have provided some information in plain language (not recommened), however, the reviewer finds it really hard to follow, see below:  

What is the input? What is the output? What is the optimization objective? What is the optimization details? How does the whole iteration/optimization start and end? Any other scientific details? 

The reviewer strongly recommends the authors to change the flowchart from plain language to pseudo code and provide all the details mentioned above in a mathematical form

4) In the experimental steps, the authors have also listed the calibration program in 7 steps (L210-227). Does this content overlap with the flowchart? It should be fine to place the steps in this section using plain language after changing the flowchart to pseudo code.  

5) In the experiments, the authors have provided mean and standard deviation of translational and rotation errors. If the reviewer has understood it correctly, the errors should be calcuated between the meaurement and the ground truth, is it correct?

The authors seems not provide such measurements and the ground truth in this experiment. Please provide more details of the conducted experiment (in a table format).

6) The literature review is not very thorough. The authors provide extensive references in extrinsic calibration of 2D LRD and cameras based on target-based and non-target approaches.

However, the recent literatures about using deep learning approach in calibration process are not completed. In addition, the literatures about leveraging additional sensors to optimize the calibration process are also missing. In this way, the following articles should be added into the reference:

Lv, X., Wang, S., & Ye, D. (2021). CFNet: LiDAR-camera registration using calibration flow network. Sensors21(23), 8112

Zhang, X., Zeinali, Y., Story, B. A., & Rajan, D. (2019). Measurement of three-dimensional structural displacement using a hybrid inertial vision-based system. Sensors19(19), 4083. 

Other comments:

1) The reviewer has realized there are a significant usage of subjunctive form throughout the manuscript, such as "XXX can be/could be/should be/would be XXX". The usage of subjunctive form usually means, within a situation, something is anticipated or imagined to happen, which has not yet occured. The reviewer recommends the authors to make this statement direct to indicate something that is known as a fact. 

2) Consider changing the title. The reviewer has carefully reviewed the manuscript and did not located any experiments that are related to studying the efficiency of the proposed system. How did the authors claim the proposed approach is "fast'? Please clarify. 

3) Please consider highlighting the contributions as bullet points at the end of the introduction for easy readibility. 

4) There are several typos seen throughout the manuscript, such as "is the baseline of the depth camera, and b is the baseline of the depth camera" in L105-106, and "the scanline of 2D LRF" (what is the scan line, is it the same as baseline b?) in L127. Please revise them accordingly. 

5) There is no need to repeatedly start new paragraph throughout the manuscript (especially the ones just after equation), such as L115, L132, L148, and L150 (there might be some others). Please confirm and make changes.   

Author Response

(The authors gave the same response as above.)

Reviewer 3 Report

The topic of the paper "Extrinsic Calibration of 2D Laser Rangefinder and Depth Camera" is very important to the research community.

The state-of-the-art is extensively presented and the authors presented clearly the innovation aimed "use only two observations to realize the joint calibration of 2D LRF and camera". However the method has to use a special purpose calibration board, and not enough theoretical novelty is presented.   The methods used and described in section 3 are well known and there is not much of a novelty here. The paper is more related to a report than to a journal paper. Basically the use of the calibration board was the novelty, which led to the extraction of the equations and parameter fitting needed, from the line extracted from the lidar to the 3D data from the camera. The method assumes that the plane extraction from the camera is precise? What is the error?  Was taken into account in the development and results analysis?  The authors presented a comparison. It is not clear what was the data gathered for methods 13 or 23. Is the comparison fair? The experiment was the same? It is not clear. Please present the data gathered for each method. What about the other methods presented in the state-of-the-art ? Why were they not used for comparison ?   Why is considered 10 mm and 1º enough, as errors? What is the gold standard?

Author Response

(The authors gave the same response as above.)

Round 2

Reviewer 1 Report

Accepted

Reviewer 2 Report

The reviewer thanks the authors to make such changes.

The current manuscript does look better and the reviewer recommends it as acceptance.